# Assessment of HPV Knowledge and Awareness among Students and Staff at IBB University, Niger State, Nigeria: Implications for Health Education and Prevention

**DOI:** 10.3390/healthcare12060665

**Published:** 2024-03-15

**Authors:** Melvin Omone Ogbolu, Miklós Kozlovszky

**Affiliations:** 1BioTech Research Center, University Research and Innovation Center, Óbuda University, 1034 Budapest, Hungary; 2John von Neumann Faculty of Informatics, Óbuda University, 1034 Budapest, Hungary; 3Medical Device Research Group, LPDS, Institute for Computer Science and Control (SZTAKI), Hungarian Research Network (HUN-REN), 1111 Budapest, Hungary

**Keywords:** cervical cancer, HPV knowledge, awareness of HPV infection, awareness of HPV vaccine, scale adaptation, quantile regression

## Abstract

In Nigeria, statistics reveal that there is a high rate of cervical cancer among women and a significant lack of awareness surrounding Human Papillomavirus (HPV), which poses a substantial risk of HPV infection. This cross-sectional survey, conducted at Ibrahim Badamasi Babangida (IBB) University, focuses on adapting and exploring the factors that influence a 20-item scale to measure HPV knowledge, evaluating knowledge-associated patterns and HPV-associated risk factors. We examined HPV vaccination rates, infection awareness, vaccine awareness, and the impact of ethnicity on HPV knowledge. Various validated forms were adapted to measure HPV awareness and knowledge. Non-parametric tests addressed non-normality. Data were presented using median and IQR and categorical data were frequency-based. Bivariate tests (Mann–Witney, Kruskal Wallis) explored knowledge-associated factors, while quantile regression (75th percentile) examined HPV knowledge factors. Variables were considered statistically significant at *p* < 0.05. The adapted 20-item knowledge scale revealed strong reliability (Cronbach’s alpha = 0.913), ensuring internal consistency. The median knowledge score was 0, with an interquartile range (IQR) of 0–5. Our findings revealed a significant lack of awareness and knowledge about HPV; only 34.8% of the population were aware of HPV infection and 25.0% were familiar with HPV vaccination. Furthermore, ethnicity was found to be significantly associated with knowledge of HPV. This study emphasizes the necessity for targeted interventions to enhance HPV awareness, especially within specific ethnic groups. Despite a robust knowledge scale, educational initiatives such as seminars/conferences about HPV and cervical cancer remain crucial in addressing this gap, ultimately reducing HPV infection and cervical cancer risks in Nigeria.

## 1. Introduction

According to research, cervical cancer (CC) poses a substantial worldwide public health challenge, especially in low- and middle-income nations, where it continues to be a primary contributor to women’s health issues in terms of non-communicable disease (NCD) which is associated with high rates of morbidity and mortality [1]. According to the World Health Organization (WHO), CC ranks as the fourth most prevalent cancer among women on a global scale. It is estimated to result in approximately 570,000 new cases and 311,000 mortalities each year, making up a significant 85% of the total burden of CC-related deaths worldwide [2]. Although awareness regarding HPV, a risk factor for CC, has grown over time, there remains a deficiency in comprehension among some individuals in the Kingdom of Bahrain, for example, as revealed by research conducted in 2019 [3]. Limited knowledge of HPV and CC in certain communities in Nigeria have been associated with reduced participation in preventive measures, such as HPV vaccination and CC screening; this can lead individuals to overlook seeking medical attention for symptoms related to HPV or CC. As a result, it is imperative to implement interventions such as community-based campaigns, outreach programs, and the provision of precise information by healthcare providers. These efforts play a pivotal role in advancing prevention, early detection, and treatment of CC by enhancing education and raising awareness about HPV and CC [2].

HPV genotypes 16 and 18 are acknowledged as high-risk variants responsible for approximately 70% of CC cases [3]. These genotypes lead to the deregulation of the expression of HPV oncoproteins E6 and E7. This, in turn, causes the abnormal growth of cervical cells and the formation of cancerous tumors [4]. Nonetheless, previous studies have shown that not all women infected with HPV 16 and 18 will develop CC. Other risk factors, such as smoking, multiple sexual partners, high parity, use of oral contraceptive (OC) pills, a compromised immune system, and a history of sexually transmitted infections (STIs), can elevate the chances of developing CC [5,6,7].

As a means of prevention, regular screening with either a Pap smear or an HPV test is recommended to diagnose abnormal cervical cells at an early stage. Additionally, vaccines that provide protection against HPV 16 and 18 are accessible for both men and women to hinder the transmission of these high-risk genotypes [8]. Furthermore, epidemiologists have discovered that cigarette smoking is a significant risk factor for both HPV infection and CC. This association is likely attributed to the negative impact of smoking on the immune system and genetic changes in cervical cells. Studies have revealed that women who smoke are more vulnerable to HPV infection and face an elevated risk of developing CC, with particularly higher risks observed among heavy and long-term smokers [6,9,10,11]. As a result, awareness programs which include topics such as promoting smoking cessation can play a crucial role in lowering the risk of HPV infection and CC among women. Additionally, it is well-established that having multiple sexual partners is a significant risk factor for both HPV infection and CC, as it heightens the chances of contracting the virus. Research findings indicate that women who report having had a higher number of sexual partners over their lifetime have a substantially higher risk of developing CC compared to those with fewer partners [12]. Hence, advocating for safe sexual practices, which includes limiting the number of sexual partners, becomes essential in mitigating the risk of HPV infection and CC [13]. In Nigeria, CC ranks as the second most prevalent cancer among women, with approximately 14,943 new cases and 10,403 deaths reported annually [14]. HPV infection as a kind of STI is widely recognized as a major risk factor for the development of CC, while specific high-risk (hr) HPV types are responsible for approximately 99% of all CC cases [15,16].

Niger State, situated in northern Nigeria, bears a substantial burden of cervical cancer, with limited availability of preventive and early detection services [17]. Research has also shown that the utilization of Pap smear tests is not prevalent among women in Nigeria [18,19]. IBB University—a notable institution in the state—serves as a significant environment to investigate the awareness and knowledge of HPV and CC among its students and staff. Young adults, including those in university, constitute a vital demographic for implementing intervention strategies focused on preventing CC through HPV vaccination and enhancing awareness about CC and its associated risk factors. Assessing the awareness of infection and vaccines, as well as the knowledge of HPV, among university students and staff is of utmost importance for the development of effective educational programs and interventions. These initiatives aim to enhance awareness, knowledge, and preventive behaviors within this specific population. Despite the critical role that knowledge plays in preventing CC, there is a shortage of research on the awareness and understanding of HPV among university students in Niger State, Nigeria. Furthermore, there is a lack of tools for assessing HPV knowledge among students in this context. Consequently, this study adapted an instrument for measuring HPV knowledge specifically for this setting. It also aimed to examine the pattern of HPV knowledge, and the factors associated with it, among students and staff at IBB University. Additionally, this study explored the rate of HPV vaccination and assessed awareness levels about HPV infection and HPV vaccination. The insights gained from this research have the potential to influence future interventions and policies focused on enhancing HPV knowledge. By doing so, we can advance prevention and control strategies, ultimately reducing the burden of CC among universities in Niger State, Nigeria, and similar populations. Given the paucity of publications on knowledge about HPV infection and CC in Niger State, our research among students and staff at IBB University in Niger State, Nigeria, was a necessity [17]. The lack of knowledge among male students/staff about their HPV status contributes to the vulnerability of females who may contract HPV and potentially develop CC in the future.

The primary aim of this study is to adapt a tool for measuring HPV knowledge. Additionally, it aims to analyze the factors associated with HPV knowledge and assess awareness levels about HPV infection and vaccine among students, academics, and non-academic staff at IBB University in Niger State.

## 2. Materials and Methods

### 2.1. Study Design and Setting

This study was conducted as an online cross-sectional survey involving students, non-academic staff, and academic staff of Ibrahim Badamasi Babangida University in Lapai, Niger State, Nigeria. To prioritize the privacy of the participants, it was important to note that Nigerian citizens have the right to protect their data, and this right is safeguarded by the Nigerian constitution. As a result, during the process of data collection, the Nigerian Data Protection Regulation (NDPR) was implemented and adhered to [20].

#### 2.1.1. Sample Size

The estimates of the sample size are based on *Z* = 1.96, *d* = 5%, and non-response rate = 10%. The following sample size determination formula for a cross-sectional study was used:(1)$$N=\frac{\left(Z{\left(\alpha \right)}^{2}\times pq\right)}{e^{2}}$$
where:

*n* = minimum sample size.

*Z* = standard normal deviation of 95% confidence interval (CI) level; corresponds to a value of 1.96.

*p* = record of HPV infection from similar studies in northern Nigeria was (0.125) [18].

*e* = level of precision of 0.04.

*q* = 1 − *p*; 1 − 0.10 = 0.90.

*N* = 277 (approximately).

na (after adjusting for a 10% non-response rate) = 304 (approximately). 

A total of 304 participants, who completed the online surveys and met the selection criteria for this study, based on the previous studies we have conducted [1], were enrolled in this study.

#### 2.1.2. Sampling Method

To address the pressing need for an adapted scale to assess knowledge about HPV infection among both university students and staff at IBB University, we conducted this study within the university community. The survey was distributed to all the 100 level–400 level students and staff of IBB University before the commencement of an online seminar we hosted about HPV and cervical cancer-related topics and risk factors. IBB University is situated in a rural setting of the north-central region of Nigeria and has an approximate population of 7000 students, 820 non-academic staff, and 310 academic staff. The university management offers first aid treatment to students and staff in an on-campus clinic. Additionally, there is a state hospital, situated approximately 3 km away, where healthcare professionals can address cases as needed. However, there have been no educational programs about HPV in Lapai, making this research the first initiative aimed at raising awareness about HPV and CC specifically within IBB University.

#### 2.1.3. Inclusion and Exclusion Criteria

The inclusion criteria for our investigation encompassed the following characteristics:i.Individuals aged 18–65 years.ii.Those with or without a history of sexual intercourse.iii.Individuals identifying as male or female only.iv.Students and staff (academic and non-academic) at IBB University.v.The three major ethnic groups (Igbo, Yoruba, and Hausa) in Nigeria were captured in this study, while the remaining ethnic groups were categorized as “others”.

The exclusion criteria for our investigation were applied as follows:i.Individuals under 18 years or above 65 years were not considered.ii.Those who identified as non-Nigerian citizens were excluded from our investigation.iii.Individuals identifying as neither male nor female were not included in the investigation.

### 2.2. Instrument

For data collection, we employed an online tool known as the Human Papillomavirus and Cervical Cancer Risk Assessment (HCRA) tool. The tool, designed within the Google survey form, is a modified version of our existing survey (the Human Papillomavirus Assessment Tool—HAT [1]) that was initially developed at the outset of this research in 2019. The adapted survey incorporates validated questions from existing tools (as referenced below) capable of measuring knowledge about HPV and CC. It also educates participants about HPV and CC after they send their responses. Additionally, a feature was enabled to prevent multiple responses from participants and to ensure that responses were collected specifically from Nigerians. Data collection instruments used for this study consisted of the demographic survey and other forms to measure knowledge and awareness of HPV. The following forms were adapted to measure the knowledge and awareness of HPV: HPV Infection Awareness Questions [21], HPV Infection Knowledge Questions (Scale) [21,22], HPV Vaccine Knowledge Questions (Scale) [21,23] and Cervical Cancer Awareness on OC Pills [24,25,26]. The data instrument was adapted for use in the study settings. The questions contained in the form can be found in Appendix A.

### 2.3. Data Analysis

#### 2.3.1. Reliability and Normality

Ensuring an elevated level of reliability in our survey was essential to guarantee the accuracy and validity of our findings. This reliability instills confidence in our ability to analyze participants’ responses and evaluate their knowledge about HPV. We assessed the survey’s reliability and coherence by examining its strong internal consistency, revealed by a value of 0.9 or higher (i.e., ≥0.9). This, in turn, strengthens the content validity of our survey. To score the knowledge scale consisting of 20 items, we employed a specific method: for positively phrased questions, we assigned a “1” to responses indicating “strongly agree” (considered correct), while all other responses were given a “0” (considered incorrect). Conversely, for negatively phrased questions, we reversed this coding process. We then summed up all the scores for each participant, resulting in a range of scores from 0 to 20; higher scores show a better understanding of the subject matter. Furthermore, we assessed the normality of the knowledge scores using the Shapiro–Wilk test for normality.

#### 2.3.2. Descriptive Statistics and Test of Association

We used descriptive statistics, including measures such as frequency and percentage distribution, as well as summary statistics, such as the median and IQR (chosen due to the non-normal distribution of the data). These statistical methods were employed to provide a comprehensive overview of the patients’ characteristics and their level of knowledge about HPV. To investigate factors that have independent associations with HPV knowledge, we employed proper statistical tests, specifically the Mann–Whitney test and the Kruskal–Wallis test, as necessary. We considered variables to be statistically significant when the *p*-value was less than 0.05, showing a meaningful relationship between the variables and HPV knowledge.

#### 2.3.3. Quantile Analysis

In our quantile analysis, we specifically concentrated on the 0.75 quantile of knowledge distribution, which corresponds to the upper quartile or the 75th percentile. For various characteristics, we calculated coefficients along with their corresponding confidence intervals. Our findings revealed that although there were certain connections between demographic characteristics and knowledge levels, not all of these effects reached statistical significance at the 0.75 quantile. Factors such as sex, education level, and ethnicity displayed varying degrees of association with knowledge levels but did not consistently achieve statistical significance. It is important to note that all data analyses were carried out using Stata MP 16 [27].

##### Model Expression

The model expression implemented for statistical analysis is the logistic regression model [28,29], as follows:(2)$$Y_{i}=\beta_{0}^{q}+X_{1}\beta_{1}^{q}+X_{2}\beta_{2}^{q}+X_{3}\beta_{3}^{q}+\varepsilon_{i}^{q}$$
where:

$Y_{i}$ is the probability of HPV knowledge up to *i*th participant.

*q* is the specific quantile associated with the equation (in this model, 0.75).

$X_{1}\;$ denotes sex, $X_{2}\;$ denotes education, and $X_{3}\;$ denotes ethnic group.

$\beta_{0}$ is the model intercept, while $\beta_{1}$, $\beta_{2}$, and $\beta_{3}$ denote the coefficients for sex, education, and ethnic group, respectively. 

$\varepsilon_{1}$ (error term) is the variance between the true value and the observed outcome (response value of students, academics, and non-academic staff) within the entire population.

## 3. Results

### 3.1. Reliability and Normality Test

We conducted a reliability test on the adapted scale and the survey used in this study to measure the participants’ knowledge of HPV, as shown in Table 1 below. This test referenced Cronbach’s alpha coefficient, a widely recognized measure of internal consistency [30]. A Cronbach’s alpha coefficient of 0.913 was found in this study, which signifies that the instrument used shows strong internal consistency in the study setting. This implies that the 20 items contained in the questionnaire are highly interconnected and collectively serve as a dependable measure of participants’ knowledge regarding HPV. The elevated level of internal consistency shows that these items consistently measure the same underlying aspects of HPV knowledge. Cronbach’s alpha value exceeding 0.9 is notably high, affirming that the questionnaire items serve as reliable indicators of participants’ knowledge. This underscores the questionnaire’s effectiveness in capturing the intended information and underscores its ability to provide valid results for the analysis of participants’ comprehension of HPV-related concepts. Furthermore, the results presented in Table 2 confirm that the knowledge score did not follow a normal distribution, as revealed by the *p*-value, *p* < 0.05.

### 3.2. Demographic Characteristics of Participants

The survey included 304 participants with diverse demographic characteristics. The median age was 28 and IQR = 24–35. The participants consisted of 44.4% females and 55.6% males. In terms of highest educational level, 76.0% had a bachelor’s degree, 18.1% had obtained a post-tertiary level of education, and 5.9% had completed high school. Employment status showed 35.2% employed, 29.3% self-employed, and 35.5% unemployed. Marital status varied: 58.2% were single, 37.2% were married, and there were smaller percentages for other categories. Ethnicity included 17.4% Hausa, 9.2% Igbo, and 21.7% Yoruba and people of other ethnic groups. In terms of sexual practices, about three-quarters of participants (66.1%) reported having had sexual intercourse. Of these, 51.3% practiced protected sex, 91.8% engaged in vaginal sex exclusively, and 8.2% practiced anal, oral, and vaginal sex (Table 3).

### 3.3. Awareness of HPV Infection and Vaccine

Table 4 presents data on participants’ knowledge of HPV infection and vaccination. Among the total respondents, 34.8% were aware of HPV infection, while 67.8% had no knowledge of HPV infection. In terms of the sources of information about HPV infection, the most mentioned source was hospitals, where participants received information from doctors, nurses, or healthcare providers, which is the most reliable means. This source was relied upon by 15.8%, while the remaining participants who have sought information about HPV infection through another medium were about 50.0%. The table also explores participants’ awareness of the HPV vaccine; 25.0% had heard of the HPV vaccine, while 75.0% reported no awareness of it. Lastly, participants were asked whether they had received the HPV vaccine. Out of the total respondents, only 3.3% reported that they had received the vaccine, while 96.7% had not received the vaccine.

### 3.4. Distribution of Participants’ Responses to HPV Infection Knowledge Questions

Analysis revealed the responses of participants regarding their knowledge about HPV infection, as shown in Table 5. A significant majority (54.2%) strongly agreed that HPV is a virus and not a bacterium. There were misunderstandings about HPV and its relation to gender, as 0.8% strongly agreed that women cannot contract HPV. The understanding of high-risk (hr) HPV genotypes varied among the participants, with 17.5% agreeing that HPV 16 and 18 are considered high-risk (hr) types. Opinions differed regarding whether HPV can be transmitted through sexual intercourse (where only 44.2% agreed). Participants’ knowledge on whether the genital warts related to HPV are cancerous yielded a low response rate (where only 37.5% agreed). There were also differing views on preventive measures, with 35.0% agreeing that using condoms can reduce the risk of contracting HPV. Overall, the responses showed a range of perspectives among the population and highlighted certain misconceptions among the participants.

### 3.5. Scored 20-Item Knowledge Scale about HPV Infection

Table 6 provides an overview of the participants’ knowledge regarding HPV infection. In summary, the median knowledge score was 0, with an IQR spanning from 0 to 5. Approximately 26.0% participants correctly recognized that women could contract HPV, 74.0% believed otherwise; 20.1% acknowledged that men can contract HPV, while 79.9% held incorrect beliefs on this. Only 6.3% correctly acknowledged that HPV infection may not cause symptoms in most cases, while most of the population, 93.7%, believed that symptoms are always present. The understanding of high-risk (hr) HPV genotypes revealed that 6.9% correctly indicated thar HPV 16 and 18 genotypes are high-risk (hr) types, while 93.1% had this knowledge statement wrong. Similarly, only 3.6% correctly recognized that HPV genotypes 6, 11, 42, and 61 are not high-risk (hr) types, while 96.4% held the incorrect belief.

The impact of sexual behavior on HPV infection was also assessed during this research. Regarding early age of sexual intercourse, 11.8% recognized that it is a risk factor for HPV, while 88.2% believed otherwise. Additionally, 14.5% correctly indicated that having multiple sexual partners increases the risk, while 85.5% held an incorrect belief. Misconceptions were observed among the population regarding the transmission and consequences of HPV infection; while 17.4% correctly acknowledged that HPV can be contracted through sexual intercourse, 82.6% held an incorrect belief on this. The participants’ understanding of HPV infection, including its persistence, treatment, and association with other conditions, varied; 2.6% correctly recognized that HPV infection usually does not require treatment to clear, while 97.4% held the opposite, incorrect, belief. Moreover, 8.9% correctly responded that HPV infection can lead to cervical cancer if left untreated, while 91.1% had this knowledge statement incorrect. Similarly, 8.9% correctly recognized that HPV infection does not cause HIV/AIDS, while 91.1% held an incorrect belief on this. The table also assessed participants’ knowledge of preventive measures; only 13.8% correctly responded that the use of condoms reduces the risk of HPV infection, while 86.2% believed otherwise.

### 3.6. Participants’ Knowledge Scores Stratified by Participants’ Characteristics

Table 7 provides an overview of the median knowledge scores and IQR, which is categorized by the participants’ profiles. The overall median score was 0, with an IQR spanning from 0 to 5. When stratified by sex, a statistically significant difference in knowledge scores was revealed between males and females (*p*-value = 0.000). Females had a median knowledge score of 0, with scores ranging from 0 to 6, while males had a median knowledge score of 0, with scores ranging from 0 to 4. These results suggest a slight discrepancy in knowledge levels between males and females. Significant disparities in knowledge scores were also found based on the participants’ highest level of education (*p*-value = 0.000). Participants with a high school or secondary education had a median knowledge score of 2.5, with scores ranging from 0 to 6. In contrast, those with tertiary education (bachelor’s degree) had a median knowledge score of 5, with scores ranging from 0 to 7. Participants with post-tertiary education (master’s degree/doctorate) had a median knowledge score of 0, with scores ranging from 0 to 4. These findings suggest that education level was associated with variations in knowledge levels, with higher education generally linked to greater knowledge. Ethnicity also showed significant differences in knowledge scores (*p*-value = 0.001). Participants of Hausa ethnicity had a median knowledge score of 0, with scores ranging from 0 to 2. In contrast, participants of Igbo ethnicity had a median knowledge score of 5, with scores ranging from 0 to 8. Participants of Yoruba ethnicity and other ethnicities had median knowledge scores of 0, with scores ranging from 0 to 5 and from 0 to 4, respectively. These results highlight the influence of ethnicity on knowledge levels and reveal variations among different ethnic groups.

### 3.7. Multivariate Analysis for Predicting Knowledge Score

Table 8 presents the results of the analysis focusing on the 0.75 quantiles. This quantile stands for the upper quartile, or the 75th percentile, of the distribution. The coefficients, along with their corresponding 95% confidence intervals (CI) and *p*-values, are reported for distinctive characteristics. Regarding sex, the coefficient for males was −2.5, which shows a negative association with the outcome variable. However, a 95% CI (−5.34, 0.34) includes zero, suggesting that the effect was not statistically significant in the quantile model Q0.75 (*p* = 0.084). Looking at education, individuals with post-tertiary education have a coefficient of 2.5, which shows a positive association. However, a 95% CI (−0.95, 5.95) includes zero, suggesting that the effect was not statistically significant Q0.75 (*p* = 0.154) in the quantile model. Similarly, for individuals with tertiary education, the coefficient was −0.5, suggesting a negative association, but the effect was not statistically significant as shown by the CI (−4.81, 3.81) and Q0.75 (*p* = 0.820). Analysis of the ethnic groups reveals that individuals from the Igbo ethnic group show a coefficient of 4, indicating a positive association. The 95% CI (0.20, 7.80) does not include zero, which shows a statistically significant effect Q0.75 (*p* = 0.039). On the other hand, the coefficients for the Yoruba ethnic group (0.5) and other ethnic groups (2.0) suggest positive associations in the quantile model, but their 95% CIs include zero, suggesting no statistically significant effects Q0.75 (*p* = 0.745) and Q0.75 (*p* = 0.232), respectively.

## 4. Discussion

The aim of this study is to evaluate through survey distribution the knowledge levels of a cohort of 304 participants about HPV infection. The survey encompassed a wide range of participants characterized by diverse demographic backgrounds, thereby enhancing a thorough and comprehensive analysis of the study’s findings. These results served to improve the participants’ comprehension of various aspects related to HPV, including its classification as a virus, its modes of transmission, the manifestation of its symptoms, identification of high-risk (hr) genotypes, recognition of potential consequences, understanding of available treatments, and awareness of preventive measures.

### 4.1. Normality and Reliability

The reliability of data is a crucial aspect of any research study, as it ensures that the measurements used are consistent and dependable. In the context of this study, a reliability test was conducted to assess the questionnaire’s internal consistency, specifically evaluating participants’ knowledge about HPV. Cronbach’s alpha coefficient, a widely utilized measure of internal consistency to ascertain the usefulness of the instrument items, was employed for this purpose [31]. The obtained Cronbach’s alpha coefficient of 0.913 indicates that the instrument used in this study exhibits excellent internal consistency. The high value of Cronbach’s alpha suggests that the items included in the questionnaire are strongly correlated and collectively form a reliable measure of participants’ knowledge about HPV. In other words, the questionnaire items consistently measure the same underlying construct of HPV knowledge. A Cronbach’s alpha value above 0.9 is considered exceptionally strong [30], indicating that the survey used in this study is highly reliable in assessing participants’ knowledge levels. This high level of internal consistency enhances the confidence in the instrument’s ability to capture proper and consistent information about participants’ understanding of HPV-related concepts. The Shapiro-Wilk analysis was conducted to assess the normality of the age variable, which is a key aspect of this study. The aim was to understand the distribution of ages among the participants and to examine whether it followed a normal distribution. The Shapiro−Wilk test is commonly used to evaluate the normality assumption of a dataset [23]. In this analysis, the test statistic (W) was calculated to be 7.22 and the corresponding *p*-value = 0.000, indicating a significant departure from normality [31].

Given that the age variable did not follow a normal distribution, it became necessary to consider alternative measures to describe the central tendency of the data. The median, as a robust statistic, was found to be proper in this scenario. Unlike the mean, which is sensitive to extreme values and the shape of the distribution [32], the median provides a more resistant measure of central tendency that is less affected by outliers or non-normality. By using the median, we were able to summarize the distribution of ages in a way that was not influenced by the non-normality observed in the data [32]. The median age was calculated to be 28, with a concentration between 24 and 35. This information accurately represents the middle value of the age distribution and provides insight into the central tendency of the dataset, regardless of its departure from normality.

### 4.2. Demographic Characteristics

The demographic characteristics of the participants revealed interesting information about the population under study. Most participants were between the ages of 24 and 35, reflecting a relatively young population. Gender distribution was almost equal, with slightly more males than females. The educational background of the participants showed a higher proportion with a bachelor’s degree, followed by post-tertiary qualifications and high school completion. Employment status varied, with a significant fraction of the population being unemployed or self-employed. Marital status also showed variation, with a higher percentage of participants being single. Ethnicity revealed a diverse composition, with a significant percentage categorized as “other” and notable representations from Hausa, Igbo, and Yoruba ethnic groups. In terms of sexual practices, the majority practiced protected vaginal sex, while a smaller percentage engaged in a combination of anal, oral, and vaginal sex.

### 4.3. Awareness of HPV Infection and Vaccine

Most of the respondents (67.8%) reported not having heard about HPV infection. In 2021, a similar proportion was reported in a previous study conducted by researchers in the city of Kosovo, where they also found that the majority of participants (66.4%) had not heard about HPV infection [33]. Another study which was also carried out in 2021 reported a low percentage of prior knowledge of HPV infection and vaccination [34]; this indicates a significant gap in awareness about HPV infection. Sources of information for HPV infection such as hospitals (through doctors, nurses, or other healthcare providers) were the most mentioned source of information for HPV awareness (15.8%). The most common source of information reported among the participants is in line with one of the three most reliable sources of HPV infection reported among youths in 2022. In the study, they found that the three most reported sources for learning and staying informed on HPV infection and vaccination were school health programs, healthcare providers, and participants’ social networks [35].

In terms of HPV vaccine awareness, only 25.4% of participants reported being aware of its existence. Hospitals (through doctors, nurses, or other healthcare providers) were the most mentioned source of information (10.9%). Some researchers also reported a similar percentage of HPV vaccine awareness in their study, where they reported that the majority of participants (≥70.1%) did not have any prior knowledge concerning the HPV vaccine [33,35]. In 2023, a systematic review and meta-analysis conducted in Ethiopia also found that a low percentage of study participants had received the vaccine, although they reported a percentage that was higher than in this study. Nevertheless, it was still below half of the participants, and this might have been as a result of more sensitization on HPV vaccine in this area [36]. These findings highlight the need for improved education and awareness campaigns to bridge the gaps in knowledge about HPV infection and the HPV vaccine. Healthcare professionals play a crucial role in providing correct information and addressing individual concerns. Public health initiatives should also use other sources, such as public health materials, online resources, social media platforms, interpersonal networks, and traditional media outlets to reach a broader audience and ensure accurate information reaches the public.

To increase awareness and knowledge about HPV infection, efforts should focus on providing reliable information through various channels, addressing misconceptions, and promoting preventive measures. Similarly, for the HPV vaccine, enhanced efforts are needed to educate the public about its benefits and importance. Collaboration between healthcare providers, public health campaigns, and educational institutions is vital to disseminate accurate information, debunk myths, and address concerns surrounding HPV vaccine uptakes.

### 4.4. Knowledge of HPV Infection

The knowledge scores of participants about HPV infection were assessed through a series of knowledge statements (20 items). The findings revealed a relatively low level of accurate knowledge among participants. Only a small percentage correctly identified HPV as a virus, while the majority held this knowledge statement incorrectly. Misconceptions were observed regarding the ability of both women and men to contract HPV, with a significant percentage believing that women cannot contract the virus. Participants also displayed a lack of awareness about symptom manifestation, as the majority believed that symptoms are always present in HPV infection. Furthermore, there were misunderstandings about high-risk (hr) HPV genotypes, transmission, consequences, treatment, and the association of HPV with other conditions. This poor knowledge of HPV was consistent with the study conducted in Lagos, Nigeria, in 2010, which discovered that a high percentage of subjects had poor knowledge of HPV and cervical cancer [37]. Correspondingly, the results of this study are in line with a study that was conducted in Turkey, where it was found that the knowledge level on HPV of participants was also low [38]. These findings emphasize the need for targeted education and awareness campaigns to improve knowledge and mitigate misconceptions related to HPV infection.

### 4.5. Factors Associated with HPV Knowledge

The analysis of knowledge scores stratified by demographic characteristics provided valuable insights into the factors influencing knowledge levels. Sex was found to have a significant association with knowledge scores, with females exhibiting slightly higher median knowledge scores compared to males. This was the same as in previous studies [38], in which it was found that the average knowledge score of females was much higher than that of males. This suggests that females may have a better understanding of HPV infection than males, highlighting the importance of sex-specific educational interventions.

Education level also revealed a significant association with knowledge scores, with participants holding a higher degree of education showing higher median knowledge scores. This was to be expected, as those that are more educated are expected to be well-informed on matters concerning health due to their exposure to information. According to a study conducted in China by Junyong and others, a significant association has been discovered between levels of education and knowledge of HPV. This suggests that higher education contributes to a better understanding of HPV infection, underscoring the need for targeted educational initiatives at different educational levels [39]. Ethnicity was another significant factor influencing knowledge levels, with participants from the Igbo ethnic group having the highest median knowledge scores. The knowledge and attitudes of individuals have also been found to be associated with ethnicity by various studies in the past. Furthermore, other epidemiological studies found ethnicity to be significantly associated with knowledge of HPV [40]. This was also consistent with a study carried out in the USA in 2009, where they found that people of different races have significantly different levels of knowledge regarding HPV infection [41]. These variations among different ethnic groups highlight the importance of culturally sensitive educational programs to address specific knowledge gaps and misconceptions prevalent within each ethnic group.

## 5. Conclusions

In this study, we successfully adapted a 20-item knowledge scale, demonstrating its consistency and reliability. However, the findings indicated a low level of awareness and knowledge about HPV among participants, highlighting the importance and relevance of the topic within an academic environment. Furthermore, ethnicity was found to be significantly associated with knowledge of HPV. These results highlight the need for targeted interventions to improve awareness and knowledge about HPV, particularly among specific ethnic groups in Nigeria, through the intervention of awareness programs such as smoking cessation and safe sexual practices, which could be organized in IBB University. This research employs the quantile regression model for analysis. The outcomes of this study could be invaluable for the Niger State Ministry of Health in enhancing HPV awareness within the academic setting. Moreover, our findings can benefit healthcare sectors, private organizations, and individuals who are seeking guidance on safe sexual practices and increasing their knowledge of HPV-related matters. 

## 6. Recommendations

This study suggests the implementation of targeted educational interventions to improve knowledge about HPV infection and its vaccine. Achieving this collaboration between healthcare providers, public health campaigns, and educational institutions will be instrumental in driving efforts to improve awareness and knowledge about HPV in this context. Similarly, because this study identified ethnicity as an associated factor of HPV infection, we suggest culturally sensitive programs to address specific knowledge gaps within different ethnic groups. Overall, conducting regular monitoring and evaluation to assess the impact of interventions and refine future initiatives may help in addressing the knowledge gap and promoting preventive measures, reducing the burden of cervical cancer in this study population.

## 7. Limitations

While this study yielded significant findings, it is important to acknowledge its limitations. These limitations include the relatively small sample size and the focus on a specific state in Nigeria, which may restrict the generalizability of the results. Additionally, relying on self-reported data may introduce biases and the study did not delve into the underlying reasons for the observed lack of knowledge or awareness. Despite these limitations, the findings still hold value and contribute to our understanding of the topic.

## Figures and Tables

**Table 1 healthcare-12-00665-t001:** Reliability Statistics.

Statistics	Value
Cronbach’s Alpha	0.913
Cronbach’s Alpha Based on Standardized Items	0.913
Number of Items	20

**Table 2 healthcare-12-00665-t002:** Shapiro–Wilk Test for Normality.

Variable	N	Z	*p* > z
Age	304	7.22	0.000

**Table 3 healthcare-12-00665-t003:** Demographic Characteristics of the Study’s Participants.

Characteristics	*n* (%)	Median (25–75 p)
Total	304 (100)	
Variables		
Age		28 (24–35)
Sex		
Female	135 (44.4)	
Male	169 (55.6)	
Highest Level of Educational		
High School/Secondary Education	18 (5.9)	
Tertiary Education (bachelor’s degree)	231 (76.0)	
Post Tertiary (master’s degree/Doctorate)	55 (18.1)	
Employment Status		
Employed	107 (35.2)	
Self-Employed	89 (29.3)	
Unemployed	108 (35.5)	
Marital Status		
Divorced	1 (0.3)	
Living with Partner	7 (2.3)	
Married	113 (37.2)	
Separated	2 (0.7)	
Single	177 (58.2)	
Widowed	4 (1.3)	
Ethnicity		
Hausa	53 (17.4)	
Igbo	28 (9.2)	
Yoruba	66 (21.7)	
Other	157 (51.6)	
Ever Had Sexual Intercourse		
Yes	195 (66.1)	
No	109 (35.9)	
Protected Sex		
Yes	100 (51.3)	
No	95 (48.7)	
Kind of sex mostly engaged in		
Anal, Oral, and Vaginal Sex	16 (8.2)	
Vaginal sex only	179 (91.8)	

**Table 4 healthcare-12-00665-t004:** Awareness of HPV Infection and Vaccine.

Variable	Yes *n* (%)	No *n* (%)
Awareness of HPV infection		
Heard about HPV Infection?	98 (34.8)	206 (67.8)
Sources of Information (Multiple Options)		
Hospitals through doctors, nurses, or healthcare providers	48 (15.8)	256 (84.2)
Public health brochures, pamphlets, flyers, or posters	41 (13.5)	263 (86.5)
Self-search on the internet	43 (14.1)	261 (85.9)
Social media through Facebook or Twitter etc.	35 (11.5)	269 (88.5)
Family member(s), friend(s), or peer(s)	24 (7.9)	280 (92.1)
Tv or radio, newspapers, or magazine	9 (3.0)	295 (97.0)
HPV Vaccine Information		
Heard of HPV Vaccine?	76 (25.0)	228 (75.0)
Sources of Information (Multiple Options)		
Hospital through doctors, nurses, or other health care provider(s)	32 (10.6)	272 (89.4)
Public health brochures, pamphlets, flyers, posters, etc.	35 (11.6)	269 (88.5)
Self-search on the internet	19 (6.3)	285 (93.7)
Social media through Facebook, Instagram, or Twitter, etc.	21 (6.9)	283 (93.1)
Family member(s), friend(s), peer(s), or colleague(s)	22 (7.3)	281 (92.7)
TV, radio, newspapers, or magazines	6 (2.0)	298 (98.0)
Have you taken the Vaccine	10 (3.3)	294 (96.7)

**Table 5 healthcare-12-00665-t005:** Distribution of Participants’ Responses to HPV Infection Knowledge Questions.

Knowledge Statements	Strongly Agree		Agree		Neutral		Disagree		Strongly Disagree	
	*n*	%	*n*	%	*n*	%	*n*	%	*n*	%
HPV is a virus, not a bacterium	65	54.2	36	30.0	6	5.0	3	2.5	10	8.3
Women cannot contract HPV infection	1	0.8	5	4.2	2	1.7	33	27.5	79	65.8
Men cannot contract HPV infection	3	2.5	9	7.5	10	8.3	37	30.8	61	50.8
In most cases, HPV infection may not cause any symptoms	19	15.8	35	29.2	31	25.8	22	18.3	13	10.8
HPV 16 and 18 genotypes are high-risk HPV genotypes	21	17.5	25	20.8	65	54.2	4	3.3	5	4.2
HPV 6, 11, 42 and 61 genotypes are high-risk HPV genotypes	11	9.2	22	18.3	77	64.2	9	7.5	1	0.8
Having sex at an early age increases the risk of getting HPV infection	36	30.0	43	35.8	21	17.5	17	14.2	3	2.5
HPV infection usually does not need any treatment to clear before it goes	8	6.7	14	11.7	19	15.8	36	30.0	43	35.8
HPV can be contracted through sexual intercourse	53	44.2	51	42.5	6	5.0	1	0.8	9	7.5
HPV-related genital warts are cancerous	17	14.2	45	37.5	34	28.3	14	11.7	10	8.3
There are several types of HPV genotypes (e.g., 6, 11, 16, 18, 70, 72)	28	23.3	32	26.7	54	45.0	5	4.2	1	0.8
HPV 16 and 18 infections can lead to cervical cancer if untreated for a long time.	27	27.6	44	44.9	23	23.5	2	2.0	2	2.0
HPV infection can be contracted from an infected person through genital skin.	30	25.0	49	40.8	22	18.3	12	10.0	7	5.8
Some HPV genotype infections can cause genital warts.	31	25.8	57	47.5	25	20.8	5	4.2	2	1.7
HPV infection can cause HIV/AIDS infections	11	9.2	19	15.8	33	27.5	30	25.0	27	22.5
Having multiple sexual partners increases the risk of contracting HPV infection.	44	44.9	41	41.8	5	5.1	3	3.1	5	5.1
HPV infection can be cured with antibiotics.	7	5.83	34	28.33	40	33.33	15	12.50	3	2.50
The use of condoms reduces the risk of HPV infection.	42	35.00	54	45.00	15	12.50	6	5.00	3	2.50
A person could be HPV infected for several years and not know it.	29	24.17	69	57.50	11	9.17	8	6.67	3	2.50
Most sexually active people will get HPV infection at some point in their lifetime.	17	14.17	31	25.83	38	31.67	25	20.83	9	7.50

**Table 6 healthcare-12-00665-t006:** Scored 20-Item Knowledge Scale about HPV infection.

Knowledge Statement	Correct		Wrong	
	*n*	%	*n*	%
HPV is a virus, not a bacterium.	36	11.8	268	88.2
Women cannot contract HPV infection.	79	26.0	225	74.0
Men cannot contract HPV infection.	61	20.1	243	79.9
In most cases, HPV infection may not cause any symptoms.	19	6.3	285	93.7
HPV 16 and 18 genotypes are high-risk HPV genotypes.	21	6.9	283	93.1
HPV 6, 11, 42 and 61 genotypes are high-risk HPV genotypes.	11	3.6	293	96.4
Having sex at an early age increases the risk of getting HPV infection.	36	11.8	268	88.2
HPV infection usually doesn’t need any treatment to clear before it goes.	8	2.6	296	97.4
HPV can be contracted through sexual intercourse.	53	17.4	251	82.6
HPV-related genital warts are cancerous.	10	3.3	294	96.7
There are several types of HPV genotypes (e.g., 6, 11, 16, 18, 70, 72).	28	9.2	276	90.8
HPV 16 and 18 infections can lead to cervical cancer if untreated for a long time.	27	8.9	277	91.1
HPV infection can be contracted from an infected person through genital skin.	30	9.9	274	90.1
Some HPV genotype infections can cause genital warts.	31	10.2	273	89.8
HPV infection can cause HIV/AIDS infections.	27	8.9	277	91.1
Having multiple sexual partners increases the risk of contracting HPV infection.	44	14.5	260	85.5
HPV infection can be cured with antibiotics.	24	7.9	280	92.1
The use of condoms reduces the risk of HPV infection.	42	13.8	262	86.2
A person could be HPV infected for several years and not know it.	29	9.5	275	90.5
Most sexually active people will get HPV infection at some point in their lifetime.	17	5.6	287	94.4

**Table 7 healthcare-12-00665-t007:** Participants’ knowledge scores stratified by their characteristics.

Characteristics	Knowledge Score	*p*-Value
	Median (25–75 p)	
Sex		0.000
Female	0 (0–6)	
Male	0 (0–4)	
Highest level of Educational		0.000
High School/Secondary Education	2.5 (0–6)	
Tertiary Education (bachelor’s degree)	5 (0–7)	
Post Tertiary (master’s degree/Doctorate)	0 (0–4)	
Employment Status		0.401
Employed	0 (0–6)	
Self-Employed	0 (0–5)	
Unemployed	0 (0–5)	
Ever Had Sexual Intercourse		0.073
Yes	0 (0–6)	
No	0 (0–3)	
Protected Sex		0.461
Yes	0 (0–6)	
No	0 (0–6)	
Ethnicity		0.001
Hausa	0 (0–2)	
Igbo	5 (0–8)	
Yoruba	0 (0–5)	
Other	0 (0–4)	

**Table 8 healthcare-12-00665-t008:** Multivariate analysis for predicting knowledge score.

Characteristics	Coefficient	95% CI Lower Limit	95% CI Upper Limit	*p*-Value
Sex				
Male	−2.5	−5.34	0.34	0.084
Education				
Post Tertiary	2.5	−0.95	5.95	0.154
Tertiary Education	−0.5	−4.81	3.81	0.82
Ethnic				
Igbo	4.0	0.20	7.80	0.039
Yoruba	0.5	−2.53	3.53	0.745
Other	2.0	−1.29	5.29	0.232

## Data Availability

The HPV and CC data collected during this project and processed for this research publication as part of this study were submitted to the Niger State Ministry of Health Ethical Review Committee (NSMOH ERC) and are available upon request.

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
