# Peer review of "Assessment of HPV Knowledge and Awareness among Students and Staff at IBB University, Niger State, Nigeria: Implications for Health Education and Prevention"

_healthcare, 2024, doi:10.3390/healthcare12060665_

Round 1

Reviewer 1 Report

Comments and Suggestions for Authors

This is a cross-sectional study to validate an HPV knowledge tool and assess HPV knowledge and awareness among students and staff at IBB University, Niger State, Nigeria.

The manuscript is well written; The methods make it possible to assess knowledge and awareness about HPV among students and employees at IBB University. However, the validation stage and some sections of the manuscript require revision.

The process of adapting and validating the scales was not adequately described. The instrument presented in Appendix A is quite different from those presented in references 21-23. (HPV Infection Awareness Questions, HPV Infection Knowledge Questions; HPV Vaccine Knowledge Questions; lines 166 and 167). As this is a study whose one of the objectives is the validation of the instrument, this stage needs to be better described.

Part of the scale appears to have a measurement model defined a priori (distribution of items by factor). Therefore, the confirmatory strategy is the most appropriate to investigate the factorial and convergent validity and the invariance of the scale model.

Therefore, it was expected that the study would present, in addition to Cronbach's Alpha, other results of psychometric sensitivity, factorial validity, convergent validity and factorial invariance.

Therefore, I suggest reviewing the methods to present the results of psychometric sensitivity, factorial validity, convergent validity, and factorial invariance.

Several information about the methods in the article are not described sufficiently for the reader to understand. For example, the inclusion and exclusion criteria were not presented clearly and objectively. To understand them, you need to go to another publication in the group (Reference 1).

It is noteworthy that the methods of a scientific study need to provide the information necessary for its understanding and enable its critical evaluation regarding possible selection and information biases.

Therefore, it is suggested to review item 2. (Materials and Methods) to present all the information necessary to understand the manuscript.

Reviewer 2 Report

Comments and Suggestions for Authors

Abstract:

Parts of the abstract need to be re-worded and edited for more streamlined language.

The level of detail of the analysis plan is not needed.

Was there any assessment of CC screening methods?

Introduction:

Overall, I think the Introduction needs some work on being re-ordered so that it can flow more smoothly. There is currently duplicated information in a couple places and it’s quite choppy.

P. 1, line 44: Please elaborate on what is meant by “specific individuals.”

P. 2, Lines 63-69: smoking is mentioned on line 57 and then further elaborated here so perhaps smoking can be removed from the single instance above and only mentioned where details are provided.

P. 2, lines 68-69; lines 74-76: The sentence about smoking cessation programs here seems unnecessary unless your paper is going to talk about such a program. This would make more sense in the Conclusions under possible next steps. This applies to the mention of advocating for safe sexual practices as well.

P. 2, lines 77-83: It seems like this information may better be suited at the beginning of the first paragraph to introduce the setting for the topic. Then talk about the specific barriers.

P. 3, Lines 104-105: “In the paucity of publications on knowledge about HPV 104 infection among students at IBB University in Niger State, Nigeria, there is a potential risk 105 of HPV transmission within this population.” – this statement seems a bit far-fetched. I think it might be better to just note that your study is needed due to the paucity of publications in this area.

P. 3., Lines 107-110: This line is an exact repeat of the statement preceding it.

P. 3, Lines 115-119: I don’t think these statements are relevant to the Introduction section.

Materials and Methods:

Section 2.1: More information is needed on the demographics of the target population – how many students and staff are there at IBB? Is it located in an urban or rural area? Do students and staff have access to healthcare on campus or nearby?

Section 2.1.1: A lot of this section was extremely technical and I feel like it could have been summarized in narrative form, rather than written as a mathematical formula.

Section 2.1.2: This seems more appropriate under the description of Instrument.

Section 2.2: I am confused as to how these scales are related to the adapted HAT and HRCA tool mentioned in the section above. Please re-work these two sections so that it’s more clear.

P. 4., Lines 200-204: These statements should be in the Results section

Are there any items related to past history of cervical cancer screening or awareness of methods? I see questions related to this in the supplementary file, but it is not clearly stated within the manuscript.

Results:

Perhaps the demographic characteristics of the sample can be presented first before the validity of the survey.

Section 3.3: I don’t know that all of that information is needed for the narrative section as it is literally repeating what is in Table 2.0. Perhaps just a summary that highlights the most important points is more suitable.

Table 3.0: To make this table more compact, consider combining Strongly Agree/Agree and Strongly Disagree/Disagree responses.

P. 8, Lines 289-P. 9, Line 304: I think this paragraph has some of the most important information; however, because it is so dense it was hard for me to pick apart what was really important and relevant. Please just summarize.

In general, a lot of the Results narrative sections needs to be summarized and shortened. It becomes too dense for the reader and hard to follow.

Discussion:

Again, I found this section to be extremely wordy and hard to follow. I think a lot of the information can be condensed. A lot of this seems to be repeated from what was stated in the results section.

Are there any other surveys among similar populations to which this information can be compared?

Comments on the Quality of English Language

Overall, this paper was well written - just tends to be very wordy and so it causes the reader to get lost and lose focus. Please summarize sections and highlight important points.

Reviewer 3 Report

Comments and Suggestions for Authors

Dear authors,

Thank you for the opportunity to review the manuscript "Validation of HPV Knowledge Tool and Assessment of HPV 2 Knowledge and Awareness Among Students and Staff at IBB 3 University, Niger State, Nigeria: Implications for Health Education and Prevention”.

The main objective of this study, i.e. the construction and validation of a tool to measure HPV awareness, is particularly timely and interesting especially in the cultural context in which the study was conducted. Furthermore, the introduction of the paper and the references given support the aim of the research and contribute to a clear understanding of it.

Some annotations: 

- in section 1: The introduction may end with the research objective supported by the relevant references given. The sentence "This research employs the quantile regression model for analysis" could be inserted in the "methods" section and the paragraph from "the outcomes" to "matters" could be included in the discussion or conclusion.

- in section 3: The validation process of psychometric properties of the HPV Knowledge Tool is limited to the evaluation of internal consistency with Cronbach's alpha. Despite the value being very high, it would have been more useful and comprehensive to explore the latent structure of the scale and discuss the communality by individual items - this would have allowed a greater understanding of the structure of the scale as well as the consistency of the items around the factor. 

The validation process does not explore the divergent and discriminant validity of the tool. This could be an important limitation of the study, which is not specified in “Section 7”. The absence of specific statistical analyses on the tool's validation process (Exploratory Factor Analysis, Confirmatory Factor Analysis, Divergent and Convergent Validity Analysis) are issues to be further investigated and, at the very least, reported as limitations in the dedicated section.

Despite the statistical limitations of the validation process described, the absence of specific awareness of the HPV detected in a large part of the sample indicates the importance and topicality of the theme as well as the need to continue to investigate the issue through the design of specific interventions.

Round 2

Reviewer 1 Report

Comments and Suggestions for Authors

The authors clarified that the psychometric properties were not evaluated in this research, as a new scale was not developed in this study, they only adapted existing instruments that were validated in a previous study.

I understand the authors' point of view, however, I emphasize that this statement is not consistent with the traditional procedure for validating instruments with a model defined a priori (distribution of items by factor). To validate the adaptation of this type of instrument, psychometric evaluation is necessary (even if it has already been validated in another context). Therefore, I maintain my position in the previous opinion: it is necessary to present, in addition to Cronbach's Alpha, other results of psychometric sensitivity, factorial validity, convergent validity and factorial invariance.

Author Response

Dear Reviewer,

Thank you for your thorough review of our manuscript; we truly value your insights as they have significantly improved the quality of the manuscript.

We would like to emphasize the purpose of our manuscript, which is to analyze and present the level of HPV knowledge within the population.
We only alongside pretested the instrument and presented Cronbach's Alpha. Therefore, the results presented are contextualized accordingly. It is essential to note that in assessing a newly developed or adapted instrument's effectiveness in measuring HPV knowledge, ensuring its validity and reliability is paramount. To clarify:

  1. Validity i.e. to evaluate whether the instrument accurately measures its intended constructs. This has been proven by other authors from whom we have adapted our instrument. The value of  Cronbach's Alpha also reveals that the adapted instrument which we have used for data collection is valid for the purpose of this research, and
  2. Reliability i.e. to determine the consistency of the measurement without significant influence from measurement errors.

Considering the above, we have provided results concerning our research objectives. We would like to also mention that a different manuscript is focusing on psychometric analysis of this same topic.

While we appreciate the mention of factorial validity, convergent validity, and factorial invariance, they are not within the scope of our objectives. Our focus has been on Cronbach's Alpha, instrument reliability, and consistency to ensure the relevance and accuracy of our findings for future research and community application. Therefore, we ask for your kind understanding in the context of this manuscript.

Thank you for your consideration and understanding.

Kind regards,
The Authors

Reviewer 2 Report

Comments and Suggestions for Authors

Thank you for adequately addressing my concerns and comments - great work!

Comments on the Quality of English Language

This is much improved!

Author Response

Dear Reviewer,

Thank you for your thorough review of our manuscript; we truly value your insights as they have significantly improved the quality of the manuscript.

We have addressed the issues you raised as regards editing minor English and we have made corrections to certain words in the manuscript. However, abbreviations and words highlighted by red lines cannot be rectified because some are not English words, and they cannot be recognized by the English dictionary in MS Word. Additionally, we acknowledge that the word, like "Lapai," may not be familiar as it refers to a specific location in Niger state, Nigeria.

We appreciate your understanding and continued support.

Best regards, The Authors

Reviewer 3 Report

Comments and Suggestions for Authors

Dear authors, 

the changes made make the objective of the study clearer and the methodology clearer. Paragraph 2.2 as formulated gives a better understanding of the statistical analyses performed.

Overall, the changes made have improved the fluency of your paper. 

Thank you for considering the suggestions indicated.

Author Response

Dear Reviewer,

Thank you for your thorough review of our manuscript; we truly value your insights as they have significantly improved the quality of the manuscript.

We would like to ask if there are other concerns regarding this manuscript as you have not indicated to sign the review report. Kindly, check the uploaded document for your review.

Many thanks.

Kind regards,
The Authors